# Integrating Cognitive Dysfunction Accommodation Strategies into an HIV Prevention Session: A 2-Arm Pilot Feasibility Study

**DOI:** 10.3390/ijerph19159430

**Published:** 2022-08-01

**Authors:** Colleen Mistler, Michael Copenhaver

**Affiliations:** 1Department of Allied Health Sciences, University of Connecticut, Storrs, CT 06269, USA; michael.copenhaver@uconn.edu; 2Institute for Collaboration on Health, Intervention, and Policy (InCHIP), University of Connecticut, Storrs, CT 06269, USA

**Keywords:** HIV prevention, medication for opioid use disorder, opioids, behavioral interventions, cognitive dysfunction

## Abstract

Cognitive dysfunction is prevalent among persons on medication for opioid use disorder (MOUD). This cognitive dysfunction has been shown to reduce HIV treatment engagement and medication adherence. We investigated the impact of integrating specific behavioral strategies into an HIV prevention session to accommodate cognitive dysfunction among people on MOUD. Patients on MOUD (*n* = 20) were randomized to one of two different HIV prevention conditions. The same HIV risk reduction content was presented to both conditions; however, the experimental condition had accommodation strategies integrated into the session. Participants completed a skills checklist at pre-, post-, and 2-week follow-up to examine the level of HIV risk reduction content learned and utilized over time. Participants in the experimental condition indicated high acceptability (95%) for the accommodation strategies. These participants also demonstrated greater improvement in the ability to properly clean a syringe, from pre- to post- (*p* < 0.02) and from pre- to follow-up (*p* < 0.02) when compared to participants in the standard condition. Results from this pilot study indicate that accommodation strategies improved participants’ ability to learn, retain, and utilize risk reduction skills over time. This foundation of research indicates a promising, innovative strategy to increase the ability for persons on MOUD to engage in HIV prevention behaviors.

## 1. Introduction

There is growing research indicating that cognitive dysfunction among people on medication for opioid use disorder (MOUD) contributes to the increased risk of HIV [1]. Persons with opioid use disorder (OUD) display a specific cognitive profile that indicates dysfunction in a spectrum of domains including executive function, attention, memory, and information processing. This cognitive profile is similar to the level of dysfunction in persons with attention deficit hyperactivity disorder (ADHD) and post-traumatic brain injury (TBI) [2]. Opioid use affects brain functioning, as the chemistry of the substance disrupts neuron activation to decrease pain receptors. This disruption in normal brain functioning, associated with opioid use, has been found to damage emotional and physical response systems [3]. Consequently, persons on MOUD demonstrate reduced ability to pay attention, process/remember/recall information, and communicate [2]. This cognitive dysfunction has been identified as a barrier to recalling, processing, and utilizing HIV prevention information among patients on MOUD [4]. These barriers diminish patient’s medical autonomy and ability to navigate healthcare services [2].

Opioid-related overdoses have also been associated with cognitive dysfunction. In a recent systematic review, Winstanley et al. (2021) discussed how opioid-related overdose often causes cerebral hypoxia (oxygen deprivation in the brain) leading to a variety of cognitive deficits [5]. Specific cognitive deficits identified in persons on MOUD following an overdose event included memory and motor impairments [5]. The prevalence of cognitive dysfunction among persons on MOUD, many of whom have experience at least one overdose [6], is drastically high. A recent study showed that 67% of people on MOUD experience cognitive dysfunction [7], and that cognitive dysfunction has impacted treatment outcomes including poor linkage/retention in treatment, suboptimal medication adherence and lower engagement in HIV risk reduction behaviors [8,9].

Behavioral interventions for persons on MOUD are often integrated into drug treatment programs to improve adherence to medication, abstinence from illicit substances, HIV prevention, overdose prevention, and overall engagement in healthcare services [2,8,9,10,11,12]. Behavioral interventions focus on increasing knowledge, motivation, and skills related to addiction and risk reduction [13]. Evidence-based behavioral intervention strategies (e.g., Cognitive Behavioral Therapy, Motivational Interviewing, Contingency Management) demand substantial memory, attention, and executive functioning to accurately recall and utilize information conveyed during intervention sessions [14]. These behavioral intervention strategies also emphasize anticipating future consequences and decision making, actions that are often diminished for persons with cognitive dysfunction [14].

Specific behavioral intervention tasks that are impeded by cognitive dysfunction in the executive function domain include self-regulation and logical reasoning [15]. These skills are needed for patients to plan for future events and rationalize decisions. Deficits in the attention domain of cognitive dysfunction often limit patients’ ability to concentrate, listen, and engage in behavioral intervention sessions [16]. Patients who have limited memory struggle with recalling cues to reduce risk behavior, learning new information, and recollecting past information [17]. Cognitive deficits in information processing can limit a patient’s ability to manage complex language, interpret feedback, and understand the consequences of behaviors [15]. These impacts on common behavioral intervention techniques often limit a patient’s engagement/retention in treatment and reduce the overall benefits of drug treatment (i.e., relapse prevention, HIV prevention) [4]. Given the impact of cognitive dysfunction on behavioral intervention strategies, and rates of cognitive dysfunction among persons on MOUD, additional research is needed to determine what methods may be effective at improving MOUD patients’ ability to maximize key HIV prevention intervention outcomes.

A majority of research on cognitive dysfunction and its impact on treatment outcomes has been conducted on persons with attention deficit hyperactivity disorder (ADHD), traumatic brain injury (TBI), and dementia [14]. Based on this empirical literature, various accommodation strategies are expected to demonstrate similar outcomes (e.g., medication adherence, retention in treatment, HIV risk reduction skills) when integrated into HIV prevention sessions in a drug treatment setting for persons on MOUD. Therefore, we sought to investigate the impact of integrating accommodation strategies into an HIV prevention session, tailored to persons on MOUD. We hypothesized the experimental condition would demonstrate significantly greater improvements in content knowledge and risk reduction skills over time, when compared to an active control condition.

## 2. Materials and Methods

### 2.1. Study Design and Sample

We conducted a 2 session 2-arm pilot trial (*n* = 20) to assess the acceptability and impact of integrating accommodation strategies into an HIV prevention session in the context of a drug treatment setting. Our sample was recruited from a parent study that included 234 individuals with opioid use disorder (OUD), assessing their preference for various HIV prevention modalities in the context of drug treatment [18]. Participants in the parent study were recruited from an addiction treatment program (APT Foundation, Inc., New Haven, CT, USA) via clinic-based advertisements and flyers, word-of-mouth, and direct referral from counselors. Individuals were eligible for this parent study if they met for following inclusion criteria: (a) 18 years or older; (b) self-reported HIV-uninfected or unknown HIV status; (c) reported drug- (i.e., sharing of injection equipment) or sex-related (i.e., condomless sex) HIV risk in the past 6 months; (d) met DSM-5 criteria for OUD; (e) on medication for opioid use disorder (MOUD); and (f) able to understand, speak, and read English. Participants were excluded if they did not report any HIV risk behaviors and/or were not in treatment for OUD. Upon completing the parent study, these participants were then invited to participate in the pilot trial. Participants from the parent study were only eligible for the pilot trial if their Brief Inventory of Neurocognitive Impairment (BINI) score indicated mild to moderate cognitive dysfunction and they were still active in drug treatment at the APT Foundation, Inc. All participants continued to receive standard of care for OUD, consisting of evidence-based biomedical and behavioral interventions for substance use disorders.

Participants were assigned to one of two different HIV prevention conditions; BINI scores, gender, age, and racial/ethnic identity were taken into consideration when assigning condition to maintain homogeneity between groups. The same HIV risk reduction content was presented in both conditions in a 45-min session; however, only the experimental condition had accommodation strategies integrated into the session (Table 1). These accommodation strategies included a written agenda, mindfulness meditation, use of white board, handouts, hands-on demonstrations, videos, and a formal closure (session summary). The strategies were selected based on a prior narrative review, as well as focus group research among patients and treatment providers in the same drug treatment setting [14,19]. Participants completed a HIV risk reduction skills checklist at pre-, post-, and 2-week follow-up, which allowed researchers to compare the two conditions in terms of the level of intervention content retained and utilized over time. Immediately following the session, participants in the experimental condition completed a survey regarding the acceptability of the accommodation strategies.

The study protocol was approved by the Investigational Review Board (IRB) at the University of Connecticut and received board approval from the APT Foundation, Inc. All participants were provided a verbal and written description of the study and were asked to sign an informed consent form prior to participation. All sessions and data collection were audio-recorded. Participants were reimbursed USD 25 for participating in the initial 45-min HIV prevention session, including the pre- and post-assessments. Participants were reimbursed an additional USD 25 for participating in the 2-week follow up assessment.

### 2.2. Assessment Tools

#### 2.2.1. Demographics

Participants were asked to self-report their age, gender, ethnicity, education level, psychiatric visitation in past 12 months, Methadone dosage in milligrams, length of substance use, and history of injection drug use (Table 2).

#### 2.2.2. Brief Inventory of Neurocognitive Impairment (BINI)

The BINI is a 57-item self-report measure designed to assess neurocognitive dysfunction among high-risk drug users enrolled in treatment [7,20]. The nine-factor measure includes a diverse set of factors with excellent to good reliability (i.e., F1 α  =  0.97 to F9 α  =  0.73) ranging from generalized neurocognitive symptoms (Global Impairment) to more specific forms of impairment (Learning-related; Language-related; Memory-related; Psychomotor/Physical; Psychomotor/Perceptual; Anger-related; Pain-associated; Traumatic Head Injury-related). Given its ease of administration, sound psychometric properties, and straightforward interpretation, the BINI is designed to serve as an abbreviated instrument to screen for neurocognitive dysfunction among patients entering or enrolled in addiction treatment and for monitoring symptoms of dysfunction over time [7,20].

#### 2.2.3. Accommodation Strategy Acceptability Survey

A 6-item survey was used to measure the acceptability of the accommodation strategies in the experimental condition. Participants indicated which strategies were most useful in helping remember, recall, and apply the information presented to them, and how these strategies were useful in the context of anticipating future behaviors (Table 3). Participants completed this acceptability survey using Qualtrics survey software [21].

#### 2.2.4. Skills Assessment

Participants’ HIV risk reduction behavioral skills were assessed as in prior randomized controlled trials [22,23,24] by having participants demonstrate the 18 steps necessary to properly clean a syringe (α = 0.80). Ratings of audio-taped demonstrations of these procedures by staff who are blind to treatment assignment have shown high inter-rater reliability in similar prior trials (inter-rater reliability = 0.98) [23]. A total score was calculated for the risk reduction skills assessment; a higher total score indicated greater ability to accurately engage in the risk reduction behavior of cleaning a syringe.

### 2.3. Data Analysis

All data analyses were conducted using SPSS v. 27 (IBM Corporation, Armonk, NY, USA) [25]. Totals and percentages of the acceptability ratings were calculated and presented to determine the extent to which the participants supported the use of the accommodation strategies to improve their ability to learn intervention content. *T*-tests and Chi-squared tests were conducted to determine if there were any significant differences between conditions on the outcome variable, in relation to all demographic variables. The outcome variable was a numerical value of how accurate participants demonstrated cleaning a syringe. Between subjects, ANOVAs were used to examine the differences in intervention content that was retained between conditions, and the differences in utilization of risk reduction skills between conditions, via the HIV risk reduction skills assessment, at the three different time points. Specifically, we investigated if there were differences between the categorical independent variable of condition (experimental vs. standard) on the numerical accuracy score of cleaning a syringe.

## 3. Results

### 3.1. Demographics

A total of 20 participants on medication for opioid use disorder (MOUD) were included in this study. Two participants from each condition did not complete the 2-week follow up (Figure 1). The average age of participants was 44.7 years old. The average BINI score (indicating cognitive dysfunction) was 130. The average methadone dose was 80.8 mg and the average length of substance use was 20.7 years. A majority (90%) of the participants reported a history of injection drug use. Fifty five percent of the sample self-identified as female, and 55% of the sample also reported visiting a psychologist in the past 12 months. Half of the sample reported having a high school degree, and 35% reported post-secondary education. The sample was well diverse in regard to ethnicity, as 45% of the sample identified as Caucasian, 35% of the sample identified as African American, and the remaining 20% identified as Hispanic/Latinx. Results from *t*-tests and chi-squared identified homogeneity between conditions across all demographic variables, expect for methadone dose.

### 3.2. Acceptability of Accommodation Strategies

Participants in the experimental condition highly endorsed (95%) the accommodation strategies that were integrated into the HIV prevention session. All participants (*n* = 10) responded that it was helpful to use a multimodal presentation of information, written agenda, discussion of risky scenarios with case studies, peer feedback, and a formal closure. The use of a mindfulness meditation was noted to be helpful by 80% of the participants, and those who endorsed this accommodation strategy noted that it helped them focus on learning the material and calmed their mind so they could learn. Participants identified that a written agenda kept the group organized, helped them pay attention better, and helped then know what to expect over the course of the session. The use of case scenarios to discuss risk was also noted to help participants plan for risky situations, learn from others, and think about how to handle future situations. Group discussions were identified to help participants understand the information in a more in-depth manner. The use of a formal closure helped participants recall intervention content, keep the information organized, and pay attention to the most important concepts discussed.

### 3.3. Impact of Accommodation Strategies

Participants in the experimental condition, who received an HIV prevention session with cognitive dysfunction accommodation strategies, demonstrated greater retention of intervention content on how to properly clean a syringe (Table 4). The average increase in total score of how to clean a syringe from pre-test to post test was significantly higher (M = 9.4, S.D. = 3.2) in the experimental condition F (1, 14) = 8.22, *p* < 0.02. The average increase in total score of how to clean a syringe from pre-test to 2-week follow up was also significantly higher in the experimental condition (M = 7.6, S.D. = 4.7), F (1, 14) = 7.02, *p* < 0.02.

## 4. Discussion

This was the first study to examine the impact of integrating accommodation strategies into an HIV prevention session in the context of a drug treatment setting for persons on MOUD experiencing cognitive dysfunction. A 2-session 2-arm pilot trial was used to research the acceptability and impact of the accommodation strategies on patients’ ability to learn, retain, and utilize HIV risk reduction content. Participants in the experimental condition reported high acceptability of the accommodation strategies, noting how these methods increased their ability to pay attention, learn and apply information, anticipate future actions, and utilize HIV prevention skills. Significant differences between conditions indicated that the accommodation strategies increased participants’ ability to properly clean a syringe as an HIV prevention measure. These findings support greater exploration of how accommodation strategies can improve treatment outcomes for persons on MOUD.

This study is a part of an innovative line of research, as these accommodation strategies have yet to be tested in a large scale randomized clinical trial. Based on research in other populations, a number of potential accommodation strategies were theorized to compensate the particular domains of cognitive dysfunction that are often present in persons on MOUD [14]. We presented this foundation of research to drug treatment providers and patients on MOUD to inform the current study and narrow down which specific strategies may demonstrate the greatest efficacy in a drug treatment setting [19]. The accommodation strategies examined in this pilot study included strategies shown to compensate the specific domains of cognitive dysfunction that are often present among persons on MOUD (i.e., attention, executive functioning, memory, information processing), and that were most frequently endorsed by key informants in focus group research [19]. Outcomes from this study indicate that the accommodation strategies improved participants’ ability to learn and retain intervention content, and to accurately utilize the recalled HIV risk reduction skills over time. To further evaluate if different combinations of the accommodation strategies generate different/optimal risk reduction outcomes among persons on MOUD, we recommend that future research could use a Multiphase Optimization Strategy (MOST).

To help maximize behavioral interventions, brief cognitive assessments are also recommended for patients on MOUD to help providers make any needed adjustments to treatment based on a specific patient’s profile [14]. Cognitive assessments utilized in clinical settings for this patient populations have included elements of the NIH Toolbox Cognition Battery [26,27]. These assessments can be helpful for clinicians to gain a baseline of the patients cognitive functioning; however, cognitive screening has not frequently been utilized in forming a patient profile for persons with OUD in a clinical setting. Recently, the Substance Abuse and Mental Health Services Administration (SAMHSA) recommended reviewing each patient’s psychosocial history (e.g., residential life, family, employment status, mental health) in combination with cognitive function screening to help develop patient profiles [28]. The Brief Inventory of Neurocognitive Impairment (BINI) was developed specifically for utilization and optimization in a drug treatment setting [7]. Future research is needed to determine the efficacy of rapidly establishing each patient’s cognitive profiles, and associating accommodation strategies with the profiles to maximize treatment outcomes.

While this study is novel in testing the impact of integrating cognitive dysfunction accommodation strategies into HIV prevention sessions for people on MOUD, there are some notable limitations. Although the BINI is a widely acceptable and highly reliable measure of cognitive functioning among persons in drug treatment, it is self-report and therefore may be considered less valid than an objective measure. Additionally, given the nature of the pilot study, the sample size was limited and we recommend this study be replicated on a larger scale to determine greater causality. Nonetheless, the small sample size was taken into consideration by the researchers a priori and this study did have a similar sample size of other HIV prevention pilot trials [29,30,31]. The researchers did control for numerous demographic variables when randomizing participants; however, there was a significant difference between Methadone dose between the two conditions. While a higher Methadone dose has been related to decreases in working memory and attention in patients who just start drug treatment; Rass et al. found that there was no difference in cognitive performance risks once patients are stable on Methadone [32]. Given all of the participants in our pilot study were stable in drug treatment and stable on Methadone, this heterogeneity between conditions is unlikely to have influenced the results. Lastly, this study was conducted in an inner-city drug treatment clinic located in the northeast region on the United States, and therefore may not be generalizable to other locations domestically and/or abroad.

## 5. Conclusions

This study provides greater insight into how patients on MOUD learn and what strategies can improve these patients’ ability to recall and apply risk reduction behaviors. As over 100,000 people died from an overdose in the past 12 months [33], and injection drug use has accounted for 10% of new HIV infections in 2018 [34], innovative methods are needed to reduce the risks associated with illicit opioid use. Integrating accommodation strategies into drug treatment settings for persons on MOUD show promise in addressing the limitations of behavioral interventions by increasing patients’ ability to attend to, recall, and utilize intervention content over time.

## Figures and Tables

**Figure 1 ijerph-19-09430-f001:**
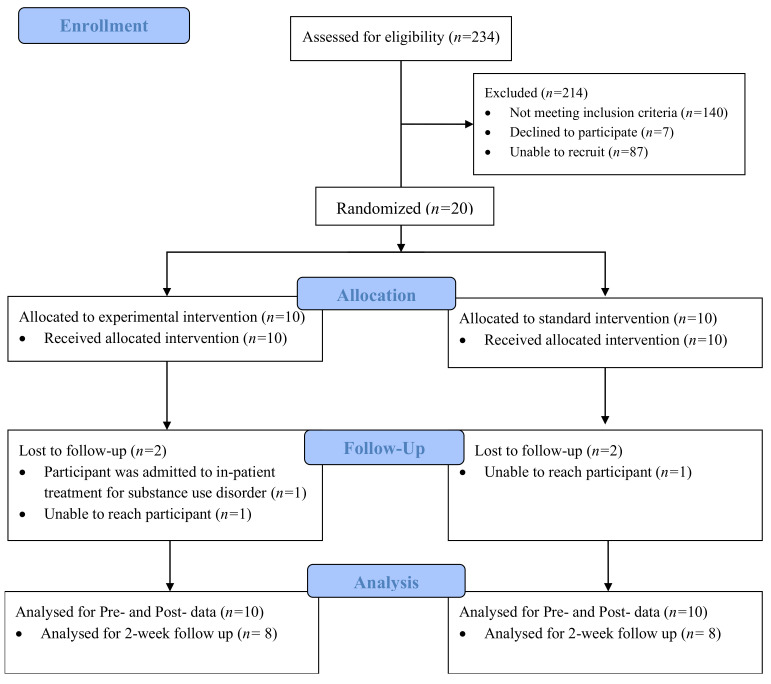
CONSORT Flow Diagram of Study Procedures.

**Table 1 ijerph-19-09430-t001:** Overview of control condition sessions and experimental condition sessions.

Time	Control Condition	Experimental Condition
0:00	**Begin Group—Introduce topics to be** **discussed** *Facilitator introduces purpose of group:* *HIV prevention and Harm Reduction*	**Begin Group—Introduce topics to be discussed** *Facilitator introduces purpose of group:* *HIV prevention and Harm Reduction* Accommodation strategy: Mention of group etiquette, use of agenda on the board for patients to refer to throughout the session and mindfulness meditation *Pre-recorded breathing exercise*
0:05	**Engaging in healthcare/advocacy**/ **resources available** *Distribute handout with list of local healthcare* *resources available*	**Engaging in healthcare/advocacy/resources available** *Distribute handout with list of local healthcare resources available * Accommodation strategy: Have patients discuss resources they have used/resources they need
0:10	**Harm Reduction** *Explain the concept of harm reduction with * *example strategies, Pyramid of Harms/types of Harm, ABC’s of harm reduction*	**Harm Reduction** *Explain the concept of harm reduction with example strategies*, *Pyramid of Harms/types of Harm, ABC’s of harm reduction* Accommodation strategy: Use of projector and use of whiteboard to relay info
0:20	**HIV infection risk/Risk behaviors/** **Risk perception** *Review how HIV is transmitted, Bodily Fluids:* *Injection Drug Use and Sex, PrEP*	**HIV infection risk/Risk behaviors/Risk perception** *Review how HIV is transmitted, Bodily Fluids:* *Injection Drug Use and Sex, PrEP* Accommodation strategy: True or False Activity, Use of Visuals, Use of Catch Phrases, Case Study Scenarios
0:25	**HIV prevention skills** *Distribute handout with the steps of how to * *apply a condom and how to clean a needle*	**HIV prevention skills** *Distribute handout with the steps of how to * *apply a condom and how to clean a needle* Accommodation strategy: Facilitator demonstration of how to apply a condom and how to clean a needle, followed by participant demonstrations
0:35	**Communication skills/refusal skills** *Group discussion on how to talk to a partner* *about safe sex, negotiating partner support*	**Communication skills/refusal skills** *Group discussion on how to talk to a partner about* *safe sex, negotiating partner support* Accommodation strategy: Video on partner communication, debrief on video, practice communication skills
0:40	**Closure** *Ask the group to share personal experiences* *and if they have any questions*	**Closure** *Ask the group to share personal experiences* *and if they have any questions* Accommodation strategy: Goal setting with patients, SMART goals
0:45	**End session**	**End session**

**Table 2 ijerph-19-09430-t002:** Demographics.

Demographic	Standard Condition (Mean, S.D.) (*n*, %)	Experimental Condition (Mean, S.D.) (*n*, %)	Total (Mean, S.D.) (*n*, %)	*p*-Value
**Age**	43 (11.05)	48 (11.52)	44.7 (11.49)	0.335
**BINI score**	125 (50.48)	135 (35.63)	130 (43.29)	0.608
**Methadone dosage (mg)**	98.1 (11.81)	63.5 (32.58)	80.8 (22.19)	0.005 *
**Length of substance use (years)**	18.1 (12.19)	23.3 (10.86)	20.7 (11.53)	0.327
**Gender**				
Male	5 (50)	4 (40)	9 (45)	0.653
Female	5 (50)	6 (60)	11 (55)	
**Ethnicity**				
Caucasian	4 (40)	5 (50)	9 (45)	0.302
African American	5 (50)	2 (20)	7 (35)	
Hispanic or Latinx	1 (10)	3 (30)	4 (20)	
**Education level**				
Some high school, no degree	1 (10)	2 (20)	3 (15)	0.663
High school degree	6 (60)	4 (40)	10 (50)	
2-year college degree	1 (10)	0 (0.0)	1 (5)	
Trade school degree	1 (10)	2 (20)	3 (15)	
Some college, no degree	1 (10)	2 (20)	3 (15)	
**Psychiatric visit in past 12 months**				
No	5 (50)	4 (40)	9 (45)	0.653
Yes	5 (50)	6 (60)	11 (55)	
**History of injection drug use**				
No	1 (10)	1 (10)	2 (10)	1.00
Yes	9 (90)	9 (90)	18 (90)	

* *p* < 0.05 is statistically significant in relation to the outcome variable; chi-squared tests were used for categorical variables (total); *t*-tests were used for numerical variables (mean, S.D.).

**Table 3 ijerph-19-09430-t003:** Acceptability of accommodation strategies.

Acceptability of Accommodation Strategies	*n*	%
Was today’s session covered verbally, visually, and with hands-on practice?	10	100%
Was it helpful to use different ways of learning?	10	100%
What method helped the most? (Check all that apply)		
Verbal	0	0%
Visual	1	10%
Hands on	1	10%
All the Above	9	90%
Was mindfulness meditation used at the start of today’s session?	10	100%
Was the mindfulness meditation helpful?	8	80%
How did mindfulness meditation help? (Check all that apply)		
Helped me focus on learning the material	4	40%
Helped calm my mind so I could learn	6	60%
Was a specific set of topics presented with an agenda at the start of today’s session?	10	100%
Was this agenda of topics helpful?	10	100%
How did this agenda help? (Check all that apply)		
Kept the group organized	3	30%
Helped me pay attention better	5	50%
Helped me know what was coming up next	5	50%
Did you discuss risk scenarios in today’s session?	10	100%
Were those discussions helpful?	10	100%
What was helpful about that discussion? (Check all that apply)		
Helped me plan for risky situations	6	60%
Helped me learn from others	1	10%
Helped me think about to handle things in the future	6	60%
Did you get feedback from other group members in today’s session?	10	100%
Did that feedback help you learn the information?	10	100%
What was useful about getting feedback (Check all that apply)		
Helped me understand the information better	6	60%
Helped me learn from others	4	40%
Did the leader close today’s group with a summary of what was covered?	10	100%
How was that summary useful? (Check all that apply)		
Helped me recall what was learned	6	60%
Helped me keep the information organized before ending	8	80%
Helped me pay attention to the overall message	1	10%
Helped me pay attention to the overall message	1	10%

*n* = total number of participants who answered “yes” to the corresponding question.

**Table 4 ijerph-19-09430-t004:** ANOVA table demonstrating increases in knowledge on how to accurately clean a syringe.

Time	Condition	Mean Difference	Std. Error	*p*-Value	95% C.I.	Observed Power
**Pre-post**	experimental	4.875	1.700	0.012 *	(1.228–8.522)	0.760
	standard	−4.875	1.700			
**Pre-follow up**	experimental	5.875	2.218	0.019 *	(1.119–10.631)	0.693
	standard	−5.975	2.218			

* *p* < 0.05 is statistically significant.

## Data Availability

Not applicable.

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
