# Peer review of "Integrating Cognitive Dysfunction Accommodation Strategies into an HIV Prevention Session: A 2-Arm Pilot Feasibility Study"

_ijerph, 2022, doi:10.3390/ijerph19159430_

Round 1

Reviewer 1 Report

In the method section, inclusion and exclusion criteria and also demographic characteristics (including education, sex, and …) should be specified!

What kind of medication did patients receive? During study condition and follow-up all patients received one kind and specific treatment? How about psychiatric medication?

My suggestion is to add a table presenting demographic data for both groups.

It seems the same HIV risk reduction content was presented in both conditions in a 45-minute session. However, in figure 1 (I think the table is better) there is some difference between the two groups. In addition, I suggest that the design of figure 1 can be changed and two strategies and groups present in one row.

I cannot see the brief inventory of neurocognitive impairment results!!!

in table 2 data related to the standard group (P and…) should be presented

It is not conmen to present the data and related figures or tables in the discussion section! This data should be moved to the result section!

The quality of figure 2 is not appropriate and also the figure legend should be completed

Author Response

Reviewer 1:

In the method section, inclusion and exclusion criteria and also demographic characteristics (including education, sex, and …) should be specified!

Thank you for identifying this oversight. We have specified inclusion and exclusion criteria and added a demographics section and corresponding table.

What kind of medication did patients receive? During study condition and follow-up all patients received one kind and specific treatment? How about psychiatric medication?

All patients were on methadone as medication for opioid use disorder. We have added their dosage into the demographics table. Additionally, all patients were active in treatment for their OUD at APT Foundation, INC and were being treated with evidence-based care consisting of both psychological and pharmacological interventions. We have further expanded on this in the methods section. We have also added psychiatric visits in the past year into the demographics table.

My suggestion is to add a table presenting demographic data for both groups.

We agree with this oversight and have provided a demographics table.

It seems the same HIV risk reduction content was presented in both conditions in a 45-minute session. However, in figure 1 (I think the table is better) there is some difference between the two groups. In addition, I suggest that the design of figure 1 can be changed and two strategies and groups present in one row.

Thank you for recommending this, we have made these edits and now believe table 1 is much easier to comprehend.

I cannot see the brief inventory of neurocognitive impairment results!!!

We have included the BINI scores in the demographics table and reran the data analysis to account for homogeneity between groups in terms of cognitive dysfunction.

in table 2 data related to the standard group (P and…) should be presented

We have edited this table to include additional relevant data points including homogeneity results.

It is not conmen to present the data and related figures or tables in the discussion section! This data should be moved to the result section! The quality of figure 2 is not appropriate and also the figure legend should be completed

We have chosen to remove figure 2 from the manuscript entirely, as all this information is provided in the ANOVA table.

Reviewer 2 Report

I thank the authors the opportunity to read their interesting manuscript regarding

Integrating Cognitive Dysfunction Accommodation Strategies into an HIV Prevention Session: A 2-Arm Pilot Feasibility Study

I found an interesting proposal considering a very relevant condition for treatment adherence, the cognitive performance among drug users.

I would like to mention some comments to take into consideration:

1. The term "Cognitive Dysfunction" is quite vague, it may be very general and lead to a wide range of cognitive impairments. Authors may consider instead particular spectrum of cognitive disabilities that they found in the  literature which may be closely related to drug use, and particular to treatment adherence.

Even when the tools used for cognitive assessment (BINI) have good reliability, and widely approved, authors may take an opportunity to identify that self reports are not very sensitive when assessing persons with drug abuse problems and cognitive dysfunction.

Please report participant's demographics, this is very valuable information needed to better understand possible features which may be related to the studied problem, furthermore, maybe related to possible confounders.

Authors should include results of normality tests, to comply with the assumption of normal distribution necessary for the use of ANOVA.

Authors report that they decided to use ANOVA:

"Between subjects ANOVAs were used to examine the differences in intervention content that was retained between conditions, and the 155 differences in utilization of risk reduction skills between condition". But in results section, it is not clear what they did compare, this needs to be precisely clarified in terms of dependent variables and measure levels.

Figure 1 is not a figure, it is a table.

Table 2 needs to be formatted according Journal's guidelines.

Figure 2 is wrongly mixed with the text, and badly formatted, please improve it.

Author Response

Reviewer 2

I thank the authors the opportunity to read their interesting manuscript regarding Integrating Cognitive Dysfunction Accommodation Strategies into an HIV Prevention Session: A 2-Arm Pilot Feasibility Study. I found an interesting proposal considering a very relevant condition for treatment adherence, the cognitive performance among drug users.

Thank you for taking the time to review our manuscript and providing relevant suggestions to improve the quality.

I would like to mention some comments to take into consideration:

  1. The term "Cognitive Dysfunction" is quite vague, it may be very general and lead to a wide range of cognitive impairments. Authors may consider instead particular spectrum of cognitive disabilities that they found in the literature which may be closely related to drug use, and particular to treatment adherence.

We have further defined cognitive dysfunction in the first paragraph of the introduction and went into more detail in the second paragraph on the impact of cognitive dysfunction particular to treatment adherence.

Even when the tools used for cognitive assessment (BINI) have good reliability, and widely approved, authors may take an opportunity to identify that self reports are not very sensitive when assessing persons with drug abuse problems and cognitive dysfunction.

Thank you for identifying this limitation, we have added this limitation and others into an additional paragraph in the discussion section.

Please report participant's demographics, this is very valuable information needed to better understand possible features which may be related to the studied problem, furthermore, maybe related to possible confounders.

We agree with this oversight and have provided a demographics table.

Authors should include results of normality tests, to comply with the assumption of normal distribution necessary for the use of ANOVA.

Thank you for bringing this to our attention. We ran normality tests between conditions for all demographic variables in regard to the outcome variable, and this is reported in the demographics variable.

Authors report that they decided to use ANOVA:

"Between subjects ANOVAs were used to examine the differences in intervention content that was retained between conditions, and the 155 differences in utilization of risk reduction skills between condition". But in results section, it is not clear what they did compare, this needs to be precisely clarified in terms of dependent variables and measure levels.

We have specified this by adding these sentences “The outcome variable was a numerical value of how accurate participants demonstrated cleaning a syringe” and “Specifically, we investigated if there were differences between the independent categorical variable of condition (experimental vs. standard) on the numerical accuracy score of cleaning a syringe.”

Figure 1 is not a figure, it is a table.

We have edited this figure to make it more digestible and now refer to it as a table.

Table 2 needs to be formatted according Journal's guidelines.

We have better formatted Table 2 by highlighting more specific statistics, which is now referred to as Table 4 in the manuscript.

Figure 2 is wrongly mixed with the text, and badly formatted, please improve it.

We have chosen to remove figure 2 from the manuscript entirely, as all this information is provided in the ANOVA table and/or results section.

Reviewer 3 Report

Integrating cognitive dysfunction accommodation strategies into an HIV prevention session: A 2-arm pilot feasibility study (ijerph-1774934)

The manuscript describes results from a small feasibility study (N = 20) that integrated strategies to overcome cognitive dysfunction (e.g., difficulties with attention span, information processing, memory) in people taking medication for opioid use disorder. Patients were randomized to one of two conditions. Each condition received the same HIV risk-reduction content (e.g., cleaning needles/syringes, correct condom use, negotiating with partners) but the experimental condition also utilized strategies to overcome cognitive dysfunction, including some repetition of material, strategies to ensure deeper processing of material, use of video elements, a mindfulness meditation to prepare for the session, and a review of material. Results showed that the intervention elements had high acceptability and the intervention group showed greater improvement from baseline in knowing the correct steps to clean a needle.

Strengths of the manuscript include the important, underexplored topic, and strong writing. Limitations include some aspects of the presentation of method and results.

Comments below may serve to strengthen the manuscript.

1. Since this is a randomized controlled trial (albeit a small feasibility study), it would be beneficial to report all of the elements recommended in the CONSORT guidelines.  Many of these elements are already reported but adding those that are missing would strengthen the manuscript. Alternatively, if those details are reported in the parent study, it would be helpful to cite that article.

2. Was the trial registered at ClinicalTrials.gov?

3. All participants used opioids at some point. Is it known how many used needles to inject opioids?

4. The primary outcome was correctly remembering the steps to clean a needle/syringe. The risk reduction curriculum included needle cleaning but also included correct condom use and negotiation with partners.  If efforts were made to assess knowledge in those areas, it would be helpful to report them. If not, it would be helpful to note that and briefly provide a rationale.

5. The manuscript notes that the Brief Inventory of Neurocognitive Impairment (BINI) was used to assess participants but I didn’t see the results reported. What were the mean/standard deviations for this measure?  Were the two groups equivalent on the BINI?

Author Response

Reviewer 3

Integrating cognitive dysfunction accommodation strategies into an HIV prevention session: A 2-arm pilot feasibility study (ijerph-1774934)

The manuscript describes results from a small feasibility study (N = 20) that integrated strategies to overcome cognitive dysfunction (e.g., difficulties with attention span, information processing, memory) in people taking medication for opioid use disorder. Patients were randomized to one of two conditions. Each condition received the same HIV risk-reduction content (e.g., cleaning needles/syringes, correct condom use, negotiating with partners) but the experimental condition also utilized strategies to overcome cognitive dysfunction, including some repetition of material, strategies to ensure deeper processing of material, use of video elements, a mindfulness meditation to prepare for the session, and a review of material. Results showed that the intervention elements had high acceptability and the intervention group showed greater improvement from baseline in knowing the correct steps to clean a needle.

Thank you for highlighting the value of our study and taking the time to review our manuscript.

Strengths of the manuscript include the important, underexplored topic, and strong writing. Limitations include some aspects of the presentation of method and results.

Comments below may serve to strengthen the manuscript.

  1. Since this is a randomized controlled trial (albeit a small feasibility study), it would be beneficial to report all of the elements recommended in the CONSORT guidelines.  Many of these elements are already reported but adding those that are missing would strengthen the manuscript. Alternatively, if those details are reported in the parent study, it would be helpful to cite that article.

Thank you for making reference to the CONSORT guidelines. We have specified additional aspects of the methods and results and added a CONSORT style figure to emphasize study procedures. The parent study is identified as reference number 18.

  1. Was the trial registered at ClinicalTrials.gov?

Given the small scale of this study, this trial was not registered at ClinicalTrials.gov

  1. All participants used opioids at some point. Is it known how many used needles to inject opioids?

We had added this statistic to the demographics section.

  1. The primary outcome was correctly remembering the steps to clean a needle/syringe. The risk reduction curriculum included needle cleaning but also included correct condom use and negotiation with partners.  If efforts were made to assess knowledge in those areas, it would be helpful to report them. If not, it would be helpful to note that and briefly provide a rationale.

Participants were also tested on their ability to accurately apply a condom; however, there were no significant findings for this outcome variable. For readability, we chose to not include these measures into this manuscript.

  1. The manuscript notes that the Brief Inventory of Neurocognitive Impairment (BINI) was used to assess participants but I didn’t see the results reported. What were the mean/standard deviations for this measure?  Were the two groups equivalent on the BINI?

We agree of the importance of including these statistics and have added them into the demographics table. There were no significant difference between groups on the BINI.

Round 2

Reviewer 1 Report

The authors responded to my comments properly and I think the current version is well-written and improved compared to the previous version.

I have just one question! As the authors reported in table 2, there is a significant difference between the two groups regarding the dosage of methadone! Which can affect the results!! The participants were randomly assigned to two groups? If yeas this difference should be explained in discussion or limitation.

Author Response

Thank you for identifying the importance of discussing this limitation. We added the following to the limitations section "The researchers did control for numerous demographic variables when randomizing participants; however, there was a significant difference between Methadone dose between the two conditions. While a higher Methadone dose has been related to decreases in working memory and attention in patients who just start drug treatment; Rass et al. found that there was no difference in cognitive performance risks once patients are stable on Methadone [32]. Given all of the participants in our pilot study were stable in drug treatment and stable on Methadone, this heterogeneity between conditions is unlikely to have influenced the results."

Reviewer 2 Report

I thank the authors for carefully taking into consideration all comments and suggestions.

Author Response

Thank you!

Reviewer 3 Report

Integrating cognitive dysfunction accommodation strategies into an HIV prevention session: A 2-arm pilot feasibility study (ijerph-1774934.R1)

The manuscript describes results from a small feasibility study (N = 20) that integrated strategies to overcome cognitive dysfunction (e.g., difficulties with attention span, information processing, memory) in people taking medication for opioid use disorder. Patients were randomized to one of two conditions. Each condition received the same HIV risk-reduction content (e.g., cleaning needles/syringes, correct condom use, negotiating with partners) but the experimental condition also utilized strategies to overcome cognitive dysfunction, including some repetition of material, strategies to ensure deeper processing of material, use of video elements, a mindfulness meditation to prepare for the session, and a review of material. Results showed that the intervention elements had high acceptability and the intervention group showed greater improvement from baseline in knowing the correct steps to clean a needle.

Strengths of the manuscript include the important, underexplored topic, and strong writing.

Overall, the authors have been highly responsive to the prior reviews and the manuscript has been strengthened as a result.

Author Response

Thank you!